# The Use of a High-Pressure Water-Ice Jet for Removing Worn Paint Coating in Renovation Process

**DOI:** 10.3390/ma15031168

**Published:** 2022-02-03

**Authors:** Grzegorz Chomka, Jarosław Chodór, Leon Kukiełka, Maciej Kasperowicz

**Affiliations:** 1Faculty of Mechanical Engineering, Branch of the Koszalin University of Technology in Szczecinek, Koszalin University of Technology, Waryńskiego 1 Street, 78-400 Szczecinek, Poland; grzegorz.chomka@tu.koszalin.pl; 2Faculty of Mechanical Engineering, Koszalin University of Technology, Racławicka 15-17 Street, 75-620 Koszalin, Poland; leon.kukielka@tu.koszalin.pl (L.K.); maciej.kasperowicz@tu.koszalin.pl (M.K.)

**Keywords:** high-pressure water jet, high-pressure water-ice jet, paint coating, geometric structure of the surface, base, surface efficiency of the process, renovation

## Abstract

The paper presents the results of investigations into the possibility of using ahigh-pressure water-ice jet as a new method for removing a worn-out paint coating from the surface of metal parts (including those found in means of transportation) and for preparing the base surface for the application of renovation paint coating. Experimental investigations were carried out in four stages, on flat specimens, sized S × H = 75 × 115 mm, cut from sheet metal made of various materials such as steel X5CrNi18-10, PA2 aluminium alloy and PMMA polymethyl methacrylate (plastic). In the first stage, the surfaces of the samples were subjected to observation of surface morphology under a scanning electron microscope, and surface topography (ST) measurements were made on a profilographometer. Two ST parameters were analysed in detail: the maximum height of surface roughness Sz and the arithmetic mean surface roughness Sa. Next, paint coatings were applied to the specimens as a base. In the third stage, the paint coating applied was removed by means of a high-pressure water-ice jet (HPWIJ) by changing the values of the technological parameters, i.e., water jet pressure p_w_, dry ice mass flow rate ${\dot{m}}_{L}$, distance between the sprinkler head outlet and the surface being treated (the so-called working jet length) l_2_ and spray angle κ for the following constants: the number of TS = 4 holes, water hole diameter φ = 1.2 mm and sprinkler head length L_k_ = 200 mm. Afterwards, the surface morphology was observed again and the surface topography of the specimen was investigated by measuring selected 3D parameters of the ST structure, Sz and Sa. The results of investigations into the influence of selected HPWIJ treatment parameters on the surface Q_F_ removal efficiency obtained are also presented. Univariate regression functions were developed for the mean stripping efficiency based on the following: dry ice mass flow rate ${\dot{m}}_{L}$, working jet length l_2_ and spray angle κ. Based on these functions, the values of optimal parameters were determined that allow the maximum efficiency of the process to be obtained. A 95% confidence region for the regression function was also developed. The results demonstrated that HPWIJ treatment does not interfere with the geometric structure of the base material, and they confirmed the possibility of using this treatment as an efficient method of removing a worn paint layer from bases made of various metal and plastic materials, and preparing it for applying a new layer during renovation.

## 1. Introduction

The continuous dynamic development of the means of transport requires the use of various materials in their manufacture. One of the main elements of vehicles of various types is their body, which is usually protected against corrosion with a paint layer. However, in order for the paint coating to fulfil its function, it is necessary, among other things, to properly prepare the surface of the base for the coatings to be applied. This refers both to factory and refinish coatings. During vehicle transport operations, factory paint coating may be damaged for various reasons; therefore, the preparation of the base for the application of the refinish paint coating becomes of particular importance.

Most frequently, damage to the factory paintwork in the vehicle body occurs as a result of natural wear and tear, such as erosion or ultraviolet radiation [1]. Other factors that contribute to damage of the factory paint coating include aggressive media (e.g., aqueous solutions of salts, acids, bases; tree sap; bird droppings; acid rain), which have a negative impact on the paint coating throughout the entire vehicle service life. Thermal factors causing the so-called temperature shock, and, in particular, extremely high or low temperatures, constitute yet another group [2]. However, it is a synergy of the factors mentioned above that is one of the most common causes of damage to the paint coating. Damage may also occur as a result of road traffic incidents. Damaged factory paintwork is most often replaced with refinish coatings. High demands are placed on these coatings as they must fulfil their protective and decorative functions as close as possible to those of factory coatings. Many technical and technological factors must be met for this to be possible. One of the most important requirements is the correct preparation of the base material [3,4]. The correct temperature as well as the quantity and cleanliness of the air flowing through the spray booth are also important [2].

Prior to regeneration of paintwork, a thorough assessment of the condition of the factory paintwork is carried out. The size and amount of visible damage as well as the type and origin of discoloration occurring on the car body are assessed. It is only then that a decision is made as to the method of paint stripping and preparation of the base for renovation. Old paintwork is usually removed using mechanical methods. Their advantage is high efficiency and the possibility to remove traces of corrosion occurring under the paint coating. The disadvantage of these methods is interference in the geometric structure of the base material prepared by the manufacturer. Therefore, when there is no need to interfere with the topography of the base material, chemical methods (alkaline, acid or solvent-based preparations) or thermal methods (pyrolysis furnaces) are used for paint removal. However, it is not always possible to use one of these methods, for example due to the necessity to disassemble the car body or the proximity of components that are sensitive to temperature or chemical agents. In such cases, sandblasting in hermetic chambers can be used. However, in this case, the topography of the base material is damaged after sandblasting [5,6,7]. In addition, abrasive particles often remain in the base material, and they form the so-called surface reinforcement following treatment, which leads to the formation of corrosion centres and defects in the refinish coating being applied.

In the search for surface treatment methods that do not damage the base, more and more attention is being paid to air-ice [8,9] or water-ice [10] jets. Most often, the carrier jet is air to which particles of crushed carbon dioxide ice are added. Such a tool for removing paint from aircraft is described in [11]. The authors of the study [12] performed the optimization of the cleaning process for circular and flat nozzles. They also presented experimental results related to the removal of paint from sheet metal and the removal of silicon seals from aluminum-magnesium alloys. Paper [13] describes the removal of particulate contaminants adhering to a surface, a process investigated using a dry ice blasting system. The experimental results showed that, for surface cleaning, dry ice blasting performs well, which is attributed to the collision of the dry ice particles with the contaminants. For submicron-sized contaminants, a lower temperature jet was required in order to produce a larger number of dry ice particles to enhance the removal efficiency.

Very few publications have investigated the use of a water-ice jet where carbon dioxide ice is used as an abrasive. The effectiveness of the surface treatment in removing paint layers using a high-pressure water-ice jet, using different nozzle variants, is presented in [14]. Frozen gas processing methods are also gaining importance, e.g., for machining of nuclear fuel pins [15]. Paper [16] describes the results of particle removal mechanisms in cryogenic surface cleaning. The agglomeration process of dry ice particles produced by expansion of liquid carbon dioxide was presented by the authors [17]. In the experiments, the temperatures of the jet flow and the tube wall were measured by thermocouples, and dry ice particles in the jet flow were observed by a high speed camera with a zoom lens. It was found that two stages of temperature reduction occurred in the jet flow, corresponding to the agglomeration process. It was also found that the particle size of the agglomerates increased and the particle velocity decreased with increasing tube diameter.

Processing can also involve a cryogenic jet of liquid nitrogen [18]. The possibility of using a jet of liquid nitrogen in combination with an abrasive (garnet) to cut metals and brittle materials was described in [19]. The results of subsequent studies [20] allowed the identification of areas of perspective application of cryogenic jets (surface cleaning, disinfection, cutting materials of different strength, cutting explosives, medicine).

The vast majority of tests are conducted on prototype test benches. The difference between the high-pressure water-ice jet (HPWIJ) presented in this paper and the ice jet technology, known as ice abrasive water jet (IAWJ), is the pressure and the abrasive used. The IAWJ method uses liquid nitrogen at the appropriate pressure and the abrasive is water ice. In the presented paper, the carrier jet is water and crushed carbon dioxide ice is used as the abrasive. A significant advantage of treatment with a high-pressure water-ice jet is the removal of surface layers without introducing surface tension [21]. In the literature, the most commonly reported results are the use of high-pressure waterjet for cutting [22,23] rather than cleaning, where the abrasive is usually garnet, corundum, or olivine. There are also published papers addressing the disintegration intensity of abrasives in the high-speed abrasive water jet (AWJ) cutting process [24,25,26,27,28]. New research has also been devoted to optimizing the cutting parameters of SiC-reinforced aluminum composite [29], and titanium alloy [30].

The possibility of using a high-pressure water jet without any additives seems to be safe for metal substrate materials and plastic parts. High-pressure water jet has previously been used to process such delicate materials as fish muscle and skin [31,32]. The necessary jet pressure to completely cut through the muscle and skin of rainbow trout was determined. The possibility also needs to be mentioned to utilise modern computer methods in further research in the aspect of modelling and simulating the aforementioned material processing methods. Experimental studies and numerous simulations of various other types of processing [33,34,35,36,37] support the validity of using such techniques in modelling the high-pressure water-ice jet.

High-pressure water jet treatment possesses an unquestionable advantage: after treatment, water is filtered and reused, thus protecting natural resources [38]. In order to increase the efficiency of the removal of different types of coatings, the water jet is mixed with CO_2_ dry ice particles. After treatment, the ice particles sublimate into the atmosphere and filtered water is reused. The hardness of dry ice particles is estimated at 2 on the Mohs scale, making them comparable to rock salt or calcite. These properties of dry ice increase the erosivity of the water jet, but there is no negative impact on the topography of the base material.

An experimental study was conducted to see if the application of high-pressure water-ice jet (HPWIJ) causes changes in the substrate material after paint removal. The influence of selected processing parameters (water jet pressure p_w_, dry ice mass flow rate ${\dot{m}}_{L}$, working jet length l_2_i and spray angle) on the surface removal efficiency Q_F_ was also investigated. Regression functions were developed for the average paint removal efficiency as a function of: dry ice mass flow rate ${\dot{m}}_{L}$, working jet length l_2_ and spray angle κ. Based on these functions, the values of the optimal parameters to obtain the maximum process efficiency were determined.

## 2. Materials and Methods

The experimental tests were carried out on a test bench constructed for this purpose. The prototype test stand (Figure 1) includes major components in the form of a stationary hydromonitor 3 and a sprinkler 7. Water is the medium that creates the so-called “jet stream” for the dry ice particles. Its source is the city water system. Water through valve 1 flows into cooler 2 where it is pre-cooled. The reduced temperature water is then compressed to the required pressure by the high-pressure pump 3. The maximum pressure of the water jet reached p_w_ = 57 MPa and the flow rate $\dot{V}=80$ dm^3^·min^−1^. The water jet pressure was stabilized by the control system 4. After compression, the temperature of the water jet rises and it needs to be cooled down again in the radiator 5. After the temperature has been reduced, the water jet enters the high-pressure gun 6 and the sprinkler 7. A high-pressure jet of water flowing through the sprinkler 7 creates a vacuum in the conduit 9, drawing ice particles from the reservoir 10 located in the temperature-controlled room. The ice particles sucked from the reservoir 10 into the sprinkler nozzle 7 are accelerated by the high-pressure water jet and shaped into a final water-ice jet sprayed onto the treated surface 8.

The test specimens were prepared in an identical manner. Once rectangular specimens sized S × H = 75 × 115 mm from a sheet had been cut out, they were first degreased and dried. Next, they were covered with two layers of phthalic primer with the trade name of Nobikor manufactured by Nobiles Włocławek (one layer along the specimens and the second across the specimens). Following the guidelines from the Nobikor manufacturer, the surfaces painted were seasoned for 24 h. Once the primer had dried, the specimens were additionally coated with a non-metallic base paint: phthalic carbamide enamel with the trade name of Autorenolak F manufactured by Polifarb Cieszyn (two coats at an interval of 24 h). The coat obtained was dried at ambient temperature (ca. 20 °C) for 10 days. Then, one layer of Spectral colourless paint was applied and seasoned for 7 days.

Experimental investigations were carried out to determine the impact of high-pressure water-ice jet treatment on the surface morphology and the surface topography structure (ST). It was also important to obtain reliable information on the selection of appropriate hydraulic and technological parameters of the cleaning process to ensure maximum surface paint removal efficiency while not interfering with the substrate material. Several substrate materials were selected for testing and paint coating were applied to them. The main consideration was their application in various means of transportation vehicle (cars, aeroplanes, light vehicle bodies, etc.). The mechanical properties of the materials used were also taken into account, as they determine the degree of damage to their geometric surface structure in the process of removing paint coatings with a water-ice jet (Table 1). The first material used is chromium-nickel stainless steel of the X5CrNi18-10 grade, which is hard yet flexible at the same time. The second base material tested is an aluminium alloy with magnesium and manganese (PA2 grade) admixtures. It is a soft material with a high susceptibility to surface damage. The third base material used for the paint coating was polymethyl methacrylate (PMMA), i.e., a macromolecular plastic characterised by good mechanical properties as well as high brittleness and susceptibility to scratching.

Paint stripping was carried out using a four-hole helical nozzle with a water hole diameter of φ = 1.2 mm (TS = 4 × 1.2 mm). The appropriate shaping of the water-ice jet was carried out in the sprinkler head with an experimentally pre-determined length L_k_ = 200 mm. Water jet pressure p_w_ = 20 MPa and p_w_ = 35 MPa was used during the tests. The dry ice flow rate was altered in the range ${\dot{m}}_{L}=52÷260$ kg·h^−1^ with a step of 26 kg·h^−1^. The distance between the sprinkler head outlet and the surface treated (the so-called working jet length) was l_2_ = 150 ÷ 400 mm and it was altered every 50 mm. The spray angle was changed in the range κ = 60 ÷ 90° with a step of 15°. Each test was repeated three times. Variants of the tests performed for X5CrNi18-10 steel substrate specimens are shown in Table 2 and Table 3.

An analogous study was conducted for samples with PA2 aluminum alloy substrates and polymethyl methacrylate (PMMA). After determining the central values (allowing to obtain the highest processing efficiency) for the dry ice output and the working length of the stream and the angle of spray, tests were carried out for the stream with pressure p_w_ = 25 MPa and p_w_ = 30 MPa.

Experimental investigations were carried out to determine the impact of the water-ice jet on the base material. For this reason, the macro- and microstructure of the surfaces of both untreated samples and samples after removal of the varnish coating with a high-pressure water-ice jet were evaluated. A JOEL JSM-5500LV scanning electronmicroscope was used to evaluate the surface morphology of the specimens examined. It was equipped with a computer system for recording and measurements that allows archiving the results obtained. The surface topography (ST) of the specimens was assessed on the basis of measurements made by means of a Talysurf CLI 2000 spatial profilometer manufactured by Taylor-Hobson. The measurement results recorded were processed and analysed using TalyMap software. The samples for measurement were 3 × 3 mm in size. Two hundred and one profiles were recorded during the measurement. The distance between the profiles was 15 µm. On one profile 1501 points were registered. The distance between the points of the profile was 2 µm. Each measurement was performed in single-pass mode. Measurements were made at the same sample location, i.e., the center of the sample. The maximum height of surface roughness (Sz) and the arithmetic mean surface roughness (Sa) were recorded during testing. The definitions and notation of spatial parameters are well known [39,40]. For each group of three samples, the average $\bar{z}$ mean value of Sa and Sz, as well as the standard deviation s(z) and the spread R(z) were calculated. The assessment of the quality of removing a worn-out paint coating from the surface of metal parts was also additionally verified with the use of the Kestler optical microscope—Vision Engineering Dynascope Ltd., Emmering, Deutschland.

A summary of the measurements of the arithmetic mean surface roughness Sa and the maximum surface roughness height Sz of the X5CrNi18-10 steel samples before paint coating is presented in Table 4. Table 5 presents the results of the same samples after removing the paint layer with a water-ice jet with pressure p_w_ = 20 MPa. Table 6 presents the results after the treatment with water jet at pressure p_w_ = 35.

A summary of the measurements of the arithmetic mean surface roughness Sa and the maximum surface roughness height Sz of the PA2 aluminium alloysamples before paint coating is presented in Table 7. Table 8 presents the results of the same samples after removing the varnish coating with the water-ice jet with pressure p_w_ = 20 MPa. Table 9 presents the results after the treatment with water jet at pressure p_w_ = 35 MPa.

A summary of the measurements of the arithmetic mean surface roughness Sa and the maximum surface roughness height Sz of the PMMA polymethyl methacrylate samples before paint coating is presented in Table 10. Table 11 presents the results of the same samples after removing the paint coating with a water-ice jet at pressure p_w_ = 20 MPa. Table 12 shows the results after the treatment with water jet at pressure p_w_ = 35 MPa.

Experimental tests were carried out in accordance with the static factor-selection programme shown in Figure 2. Three-fold repeatability of the tests was used. The results of the experiments are recorded in Table 13, Table 14, Table 15, Table 16, Table 17 and Table 18. The surface treatment capacity for each sample Q_F_m^2^·h^−1^ was determined during testing. The mean value of the surface treatment efficiency ${\bar{Q}}_{F}$ m^2^·h^−1^ and the standard deviation and spread were calculated for three samples treated with identical parameters.

The mean values of the object’s outputs ${\bar{Q}}_{F}$ were approximated by a third degree polynomial obtaining a regression equation as one-parameter functions:(1)$${\hat{\bar{Q}}}_{F}=b_{o}+b_{1}ࢫ\bar{x}+b_{2}ࢫ{\bar{x}}^{2}+b_{3}ࢫ{\bar{x}}^{3},\;[m^{2}\cdot h^{-1}]$$
where:

$b_{o},{\;b}_{1},b_{2}{\;\mathrm{and}\;b}_{3}$—unknown coefficients of the regression equation,

$\bar{x}$—input variables: $\bar{x}={\dot{m}}_{L}$ [kg·h^−^^1^] or $\bar{x}=l_{2}$ [mm] or $\bar{x}=\kappa$ [°] or $\bar{x}=p_{w}$ [MPa].

Using matrix calculus, the column vector **{b}** of the unknown coefficients in Equation (5) was calculated from the matrix formula:(2)$$\left\{b\right\}={\left({\left[\bar{X}\right]}^{T}\left[\bar{X}\right]\right)}^{-1}{\left[\bar{X}\right]}^{T}\left\{\bar{Y}\right\},$$
where:

$\left[\bar{X}\right]$–input variable matrix of dimension N × L. For data N = 5, 6, 7 and 9 and L = 4, the following matrix forms $\left[\bar{X}\right]$ were developed that are presented in Table 19:

${\left[\bar{X}\right]}^{T}$—transposed matrix $\left[\bar{X}\right],$

${\left({\left[\bar{X}\right]}^{T}\left[\bar{X}\right]\right)}^{-1}$—covariance matrix,

$\left\{\bar{Y}\right\}$—column vector of the average values of the experimental results (Table 13, Table 14, Table 15, Table 16, Table 17 and Table 18).

The boundaries of the confidence region for Regression Function (1) were determined from the following formula:(3)$${\hat{\bar{Q}}}_{F}\pm t_{\mathrm{kr}\left(\alpha ;f=N-L\right)}\cdot \frac{S_{R}}{\sqrt{N-L-1}}\cdot \sqrt{{\left\{\bar{x}\right\}}^{T}{\left({\left[\bar{X}\right]}^{T}\left[\bar{X}\right]\right)}^{1}\left\{\bar{x}\right\}},\;[m^{2}\cdot {h^{-}}^{1}]$$
where:

${\hat{\bar{Q}}}_{F}$—regression equation according to Formula (1),

$t_{\mathrm{kr}\left(\alpha ;f=N-L\right)}$—critical value of Student t test for significance level α = 0.05 and the number of the degrees of freedom f = N-L (here, f = 1, 2, 3 or 5),

N—number of measurement points in the experimental design N = 5, 6, 7 or 9,

L—number of unknown coefficients in Regression Equation (1); here, L = 4,

$\left\{\bar{x}\right\}$ and ${\left\{\bar{x}\right\}}^{T}$—column vector of the functions of input variables (test factors in real form) and its transposition:${\left\{\bar{x}\right\}}^{T}=\left[1\;{\bar{x}\bar{x}}^{2}{\bar{x}}^{3}\right]$,

$S_{R}=\overset{i=N}{\underset{i=1}{\sum }}{\left({\hat{\bar{y}}}_{i}-{\bar{y}}_{i}\right)}^{2}$—residual variance,

${\hat{\bar{y}}}_{i}$—average values of model outputs for plan points calculated from Equation (1) (${\bar{y}}_{i}={\hat{\bar{Q}}}_{\mathrm{Fi}}$),

${\bar{y}}_{i}-$ average values of experimental results (Table 13, Table 14, Table 15, Table 16, Table 17 and Table 18).

The test results after statistical processing according to the algorithm presented in the study [41] were used to develop regression equations. The results of single-factor tests are successive functions in the form of a third degree polynomial that defines the influence of the factor covered by the tests: the mass flow rate of dry ice, the working jet length and the spray angle on the surface average paint stripping rate as well as the 95% confidence region for the regression function (Figure 3).

The regression equations developed take the following form.

Dependence of the average surface efficiency ${\hat{\bar{Q}}}_{F}$ [m^2^·h^−1^] of the removal of the paint coating from the surface of the X5CrNi18-10 steel specimen from the dry ice mass flow rate ${\dot{m}}_{L}$ [kg·h^−1^]:(4)$${\hat{\bar{Q}}}_{F}=-1214\cdot 10^{-10}\cdot {\left({\dot{m}}_{L}\right)}^{3}+4327\cdot 10^{-8}\cdot {\left({\dot{m}}_{L}\right)}^{2}-372\cdot 10^{-5}\cdot {\dot{m}}_{L}+0.1077,\;R=0.9944,\;p_{w}=20\;[\mathrm{MPa}]$$
(5)$${\hat{\bar{Q}}}_{F}=-884\cdot 10^{-10}\cdot {\left({\dot{m}}_{L}\right)}^{3}+358\cdot 10^{-7}\cdot {\left({\dot{m}}_{L}\right)}^{2}-305\cdot 10^{-5}\cdot {\dot{m}}_{L}+0.1004,\;R=0.9982,p_{w}=35\;[\mathrm{MPa}]$$

Dependence of the average surface efficiency ${\hat{\bar{Q}}}_{F}$ [m^2^·h^−1^], of the removal of paint from the surface of the PA2aluminiumalloy specimen from the dry ice mass flow rate ${\dot{m}}_{L}$ [kg·h^−1^]:(6)$${\hat{\bar{Q}}}_{F}=-1478\cdot 10^{-10}\cdot {\left({\dot{m}}_{L}\right)}^{3}+519\cdot 10^{-7}\cdot {\left({\dot{m}}_{L}\right)}^{2}-4217\cdot 10^{-6}\cdot {\dot{m}}_{L}+0.1187,\;R=0.9942,\;p_{w}=20\;[\mathrm{MPa}]$$
(7)$${\hat{\bar{Q}}}_{F}=-878\cdot 10^{-10}\cdot {\left({\dot{m}}_{L}\right)}^{3}+35\cdot 10^{-6}\cdot {\left({\dot{m}}_{L}\right)}^{2}-242\cdot 10^{-5}\cdot {\dot{m}}_{L}+0.0661,\;R=0.9999,\;p_{w}=35\;[\mathrm{MPa}]$$

Dependence of the average surface efficiency ${\hat{\bar{Q}}}_{F}$ [m^2^·h^−1^], of the removal of the paint coating from the surface of the X5CrNi18-10 steel specimen from the working jet length $l_{2}$ [mm]:(8)$${\hat{\bar{Q}}}_{F}=54\cdot 10^{-10}\cdot {\left(l_{2}\right)}^{3}-7078\cdot 10^{-9}\cdot {\left(l_{2}\right)}^{2}+24\cdot 10^{-4}\cdot l_{2}-0.1272,\;R=0.9982,\;p_{w}=20\;[\mathrm{MPa}]$$
(9)$${\hat{\bar{Q}}}_{F}=93\cdot 10^{-10}\cdot {\left(l_{2}\right)}^{3}-10^{-5}\cdot {\left(l_{2}\right)}^{2}+315\cdot 10^{-5}\cdot l_{2}-0.1602,\;R=0.9999,\;p_{w}=35\;[\mathrm{MPa}]$$

Dependence of the average surface efficiency ${\hat{\bar{Q}}}_{F}$ [m^2^·h^−1^], of the removal of the paint coating from the surface of the PA2 aluminium alloy specimen from the working jet length $l_{2}$ [mm]:(10)$${\hat{\bar{Q}}}_{F}=94\cdot 10^{-10}\cdot {\left(l_{2}\right)}^{3}-112\cdot 10^{-7}\cdot {\left(l_{2}\right)}^{2}+367\cdot 10^{-5}\cdot l_{2}-0.1903,\;R=1,\;p_{w}=20\;[\mathrm{MPa}]$$
(11)$${\hat{\bar{Q}}}_{F}=114\cdot 10^{-10}\cdot {\left(l_{2}\right)}^{3}-127\cdot 10^{-7}\cdot {\left(l_{2}\right)}^{2}+4096\cdot 10^{-6}\cdot l_{2}-0.1912,\;R=0.9999,\;p_{w}=35\;[\mathrm{MPa}]$$

Dependence of the average surface efficiency ${\hat{\bar{Q}}}_{F}$ [m^2^·h^−1^], of the removal of the paint coating from the surface of the PA2 aluminium alloy specimen from the spray angle κ [°]:(12)$${\hat{\bar{Q}}}_{F}=-864\cdot 10^{-9}\cdot {\left(\kappa \right)}^{3}+968\cdot 10^{-7}\cdot {\left(\kappa \right)}^{2}+114\cdot 10^{-5}\cdot \kappa -0.0844,\;R=1,\;p_{w}=20\;[\mathrm{MPa}]\;$$
(13)$${\hat{\bar{Q}}}_{F}=-106\cdot 10^{-8}\cdot {\left(\kappa \right)}^{3}+12\cdot 10^{-5}\cdot {\left(\kappa \right)}^{2}+1097\cdot 10^{-6}\cdot \kappa -0.095,\;R=1,\;p_{w}=35\;[\mathrm{MPa}]\;$$

Dependence of the average surface efficiency ${\hat{\bar{Q}}}_{F}$ [m^2^·h^−1^] of the removal of the paint coating from the surface specimen from the water-ice jet pressure p_w_ [MPa]:X5CrNi18-10 steel
(14)$${\hat{\bar{Q}}}_{F}=0.001932\cdot p_{w}+0.09032,\;R=0.995$$

PA2 aluminium alloy


(15)
$${\hat{\bar{Q}}}_{F}=0.00294\cdot p_{w}+0.1154,\;R=0.995$$


polymethyl methacrylate PMMA


(16)
$${\hat{\bar{Q}}}_{F}=0.00368\cdot p_{w}+0.1368,\;R=0.995$$


## 3. Results and Discussion

### 3.1. The Surface Quality of the Base

The surface quality of the base after treatment plays an important role in the paint stripping process. In order to verify whether the High-Pressure Water Ice Jet (HPWIJ) treatment does not damage the base, the results of 3D surface topography (ST) and morphology of the surface of the specimens before and after paint stripping were compared.

An example of a scanning electron microscope SEM image of the surface morphology of an X5CrNi18-10 steel specimen before paint coating is presented in Figure 4a. It shows a characteristic fine-grained structure with clearly visible dark traces of grain boundaries. The isometric appearance (that is a topographic image) of the surface of the same steel is shown in Figure 4b. In this case, irregularly distributed protrusions and indentations were observed. As a result of the measurements carried out, the maximum height of the surface roughness was found to be Sz = 7.4 μm, while the arithmetic mean surface roughness was found to be Sa = 0.77μm.

After removal of the paint coating using a high-pressure water-ice jet, carried out with different treatment parameters (p_w_ = 20 MPa and p_w_ = 35 MPa), scanning electron microscope images of the surface were observed, an example of which is shown in Figure 5a. You can see the very fine grain structure with its characteristic dark borders. To a small extent, you can see the individual primer paint particles remaining at the grain boundaries. They are only discernible at 400× magnification. Comparing the crystalline structure of the surface of the specimen before (Figure 4a) and after treatment (Figure 5a), it was found that they are similar, no significant differences were found.

Similar conclusions were drawn with regard to the measurement of ST parameters. Here, too, almost identical results to those recorded for unpainted specimens were found. The surface topography, consisting of hills and depressions, is also almost identical. Their occurrence is rather irregular and similar to the results of specimens that are not subjected to the water-ice jet. The maximum height of the surface roughness of the specimens treated with the high-pressure water-ice jet was Sz = 7.3 μm, while the arithmetic mean surface roughness was Sa = 0.78 μm. The results obtained do not differ significantly at the adopted level of significance α = 0.05 from the results obtained for those specimens that were not painted.

Similar tests as for steel base specimens were carried out for PA2 aluminium alloy specimens. An example of a scanning image of the surface morphology and an isometric image of the surface of PA2 aluminium alloy specimens prior to paint coating application is presented in Figure 6. It was established that the surface of the PA2 aluminium alloy specimen is covered with parallel traces of unidirectional periodic structure formed during the rolling process. On the surface of the material, small scratches and spot indentations can also be seen as residues from previous machining processes. Based on the measurements of the SGS parameters, the height of the irregularities was found to be Sz = 10.6 μm. The second important ST parameter, i.e., the arithmetic mean surface roughness, for the PA2 aluminium alloy base is Sa = 1.04 μm.

Figure 7 presents the results of surface topography observations under a scanning electron microscope and of the measurements of ST parameters of PA2 aluminium alloy base material, from which the paint coating was removed by a high-pressure water-ice jet with pressure p_w_ = 35 MPa and dry ice mass flow rate ${\dot{m}}_{L}=208$ kg·h^−1^ using the distance between the sprinkler nozzle outlet, the surface treated l_2_ =250 mm and the jet spray angle κ = 90°. In the microscopic image, alternating protrusions and indentations are observed, ones which indicate the application of the base rolling operation. Fine scratches and indentations can also be seen. Analogous traces were observed for all those specimens from which paint coatings were removed when the maximum pressure of the water-ice jet p_w_ = 35 MPa was applied and with other process parameters being variable. The measurements of the ST parameters (Figure 7b) demonstrated that the height of the roughness, which is identical to previous studies for those specimens that were not painted, is Sz = 10.6 μm. Similar conclusions were drawn based on the measurements of the arithmetic mean deviation of the surface roughness. In this case, it was also observed that the results obtained with the value Sa = 1.06 μm are almost identical to those obtained for the specimens that were not painted. By analysing all the results obtained, it was found that the use of a high-pressure water-ice jet to remove paint coatings from PA2 aluminium alloy does not cause any changes in the ST parameters. There were also no significant differences in the appearance of the base morphology on the scanning images. The assessment of the quality of removing a worn-out paint coating from the surface of metal parts was also additionally verified with the use of the Kestler optical microscope—Vision Engineering Dynascope Ltd., Emmering, Deutschland.

A microscopic photograph of the surface of a specimen made of PMMA polymethyl methacrylate, at 50 times magnification, is shown in Figure 8a. This is a uniform, smooth and even surface. For specimens made of PMMA, the geometric structure of their surfaces before the application of paint coatings is shown in the example of Figure 8b.

Based on the image provided and the measurement results obtained, it was found that the surface of the specimens made of PMMA polymethyl methacrylate possesses the lowest roughness in comparison with PA2 aluminium alloy and X5CrNi18-10 steel. The arithmetic mean surface roughness of the PMMA prior to treatment was Sa = 0.49 μm, while the height of the roughness was Sz = 5.47 μm.

A scanning electron microscope image of a PMMA polymethyl methacrylate specimen, from which the paint coating was removed by means of a high-pressure water-ice jet is presented in Figure 9a. It shows clear signs of surface chipping. Similar defects are observed during ST measurements. The surface morphology contains numerous “craters”, which are the effect of the water-ice jet impact. As a result of the formation of micro-chips in the PMMA surface, the arithmetic mean deviation of its surface roughness increases to Sa = 1.48 μm. Damage to the base material as a result of the application of the water-ice jet is also evidenced by a significant increase in the height of the irregularities amounting to a maximum of as much as Sz = 38.4 μm.

Reducing the working pressure of the jet to 20 MPa and the dry ice output to a maximum of ${\dot{m}}_{L}=156$ kg·h^−1^ enabled an effective paint stripping tool to be obtained. At the same time, the applied treatment did not cause damage to the substrate surface. A scanning electron microscope image of the surface of the PMMA polymethyl methacrylate sample from which the paint layer was removed is shown in Figure 10a, while the surface topography of the substrate is shown in Figure 10b.

Based on the results obtained from the conducted research, it was concluded that the removal of paint layers with a high-pressure water-ice jet is possible without disturbing the geometric structure of the substrate material. This includes substrates made from X5CrNi18-10 steel, PA2 aluminum alloy, and polymethyl methacrylate (PMMA).

### 3.2. Influence of Selected Treatment Parameters on the Surface Performance of the HPWIJ Process

On the basis of the results obtained from the research conducted, it was established that the removal of paint coatings from various base materials is possible without disturbing the structure of the base material. However, in order to accomplish this, it is necessary to know the values of the technological parameters which enable the cleaning process to be carried out correctly. From the perspective of practical applications, for operators that employ a high-pressure water-ice jet for removing paint coatings, it is not only the effect of such technology on the base material that is of great importance but, above all, this is the impact on the surface treatment efficiency achieved during work. First, it should be noted that paint stripping with a high-pressure water jet cannot be compared to conventional abrasive blasting methods [42,43,44,45,46,47]. This is not a competitive method considering the hardness and density of the abrasive that is admixed. However, the most important advantage of machining using an ice-water jet is the absence of the traces of impact on the base material. Increasingly, manufacturing companies are looking for technologies of this type, e.g., for cleaning injection moulds and ship hulls. It is therefore important to consciously use new technologies in industry. For this purpose, it is necessary to determine the values of technological and hydraulic parameters in order to achieve the maximum treatment efficiency while ensuring that there is no destructive effect on the base.

Figure 11 presents the effect of CO_2_ dry ice particle output on the surface paint coat stripping efficiency of X5CrNi18-10 steel for jet pressures p_w_ = 20 MPa and p_w_ = 35 MPa. As the pressure of the water jet increases, its velocity increases, and so does the velocity of CO_2_ dry ice particles that are accelerated by it. The higher kinetic energy of dry ice particles results in a stronger mechanical (impact) effect on the surface being treated. The sublimation of CO_2_ particles in the treatment region is also more intense. In addition, a higher-pressure water jet exerts a greater force on the paint coating to be removed. This is why the microcracks in the top layers of the paint film are separated and torn apart more quickly. Consequently, a higher pressure water-ice jet treatment leads to an increase in the surface paint coat stripping efficiency. At the same time, it should be noted that a higher pressure water jet, with its higher kinetic energy, is able to accelerate a larger mass of dry ice to the maximum speed than a lower pressure jet.

In the case of X5CrNi18-10 steel specimens, a minimum surface paint coat stripping efficiency of ${\bar{Q}}_{F}$ = 0.014 m^2^·h^−1^ is obtained with dry ice mass flow rate ${\dot{m}}_{L}=52$ kg·h^−1^ and water jet pressure p_w_ = 20 MPa. An increase in dry ice flow rate results in a higher number of CO_2_ particles hitting the surface being cleaned. Weakened by the impact of the dry ice particles and of the water jet, the outer layer of the paint coatis removed more quickly. Doubling the CO_2_ dry ice output (${\dot{m}}_{L}=104$ kg·h^−1^) (Figure 11a) leads to a more than threefold increase in the surface paint coat stripping efficiency (${\dot{Q}}_{F}$ = 0.05 m^2^·h^−1^). The optimum dry ice particle output to be used for stripping with water-ice jet pressure p_w_ = 20 MPa is ${\dot{m}}_{L}=182$ kg·h^−1^. This being the case, surface paint coat stripping efficiency reaches the value (${\dot{Q}}_{F}$= 0.125 m^2^·h^−1^). With an increasing CO_2_ output, the decreasing number of dry ice particles to be used to remove the paint coating on the “unit surface” indicates that the “supercooled” water jet fully accelerating dry ice particles added to it is a tool with maximum erosivity. A flow rate of CO_2_ particles that is too high causes the nozzle pass to be “choked” with excess ice. The water jet is then unable to accelerate dry ice particles so that they could achieve maximum velocity. Therefore, with a flow rate being too high, dry ice particles have less kinetic energy compared to particles that are fully accelerated. The end result is a decrease in surface paint coat removal efficiency. In addition, using CO_2_ mass flow rate that is too high results in increased treatment costs, which is undesirable from a practical point of view. For example, with dry ice mass flow rate of ${\dot{m}}_{L}=208$ kg·h^−1^, surface paint coat stripping efficiency decreased to Q_F_ = 0.116 m^2^·h^−1^. Under these conditions, more than 75 CO_2_ dry ice particles have to be used to obtain 1 mm^2^ of cleaned surface.

The use of a high-pressure water-ice jet with water jet pressure p_w_ = 35 MPa (Figure 11b) ensures maximum surface paint coat stripping efficiency ${\dot{Q}}_{F}$= 0.228 m^2^·h^−1^ with dry ice flow rate ${\dot{m}}_{L}=216$ kg·h^−1^. This is an increase in the maximum surface paint coat stripping efficiency ∆${\dot{Q}}_{F}$= 0.112 m^2^·h^−1^ compared to that obtained for a jet with pressure p_w_ = 20 MPa, while maintaining the remaining processing parameters at the same level.

The effect of dry ice output on the surface stripping efficiency of paint coatings from PA2 aluminium alloy specimens for water jet pressure p_w_ = 20 MPa and p_w_ = 35 MPa is presented in Figure 12a,b. Similarly as for the specimens with X5CrNi18-10 steel base, also in this case, an increase of water jet pressure is accompanied by an increase of surface paint coat stripping efficiency. It needs to be noted that for the removal of the paint coating from PA2 aluminium alloy, it is most beneficial to use a water-ice jet with pressure p_w_ = 35 MPa and CO_2_ output ${\dot{m}}_{L}=225$ kg·h^−1^ because it is then that the highest surface treatment efficiency is obtained.

When using a jet with pressure p_w_ = 20 MPa, the best treatment effects, based on experimental research results, are obtained using dry ice flow ranging from ${\dot{m}}_{L}=156$ kg·h^−1^ to ${\dot{m}}_{L}=182$ kg·h^−1^ (Table 3). With these treatment parameters, as a result of a cumulative interaction of dry ice particles (i.e., mechanical interaction and interaction from their sublimation) and the water jet, a water-ice jet of the highest erosivity is obtained. When dry ice output ${\dot{m}}_{L}=234$ kg·h^−1^(for p_w_ = 35 MPa) (Figure 12b) and ${\dot{m}}_{L}=182$ kg·h^−1^ (for p_w_ = 20 MPa) (Figure 12a) is exceeded, the water jet is unable to accelerate the increased CO_2_ mass to its maximum velocity, the result being decreased treatment efficiency.

Figure 13 and Figure 14 show the effect of the distance between the sprinkler nozzle head outlet l_2_ (also known as the working jet length) and the surface being treated on the maximum surface efficiency Q_F_ of paint coat stripping. It should be noted that when using a working jet length that is too small, high erosivity is achieved but not the highest treatment efficiency is obtained. In the case of the base made of X5CrNi18-10 steel, an increase in the working length of the water-ice jet from l_2_ = 150 mm to l_2_ = 200 mm is accompanied by an increase in the surface paint coat stripping efficiency by 18 % on average when using a water-ice jet with pressure p_w_ = 20 MPa (Figure 13a). The maximum increase in stripping efficiency of just under 26% is achieved by changing the working jet length from l_2_ = 150 mm to l_2_ = 250 mm. Any further increase in the working jet length from l_2_ = 150 mm to l_2_ = 300 mm only results in a 9% increase in stripping efficiency. The use of length l_2_ = 350 mm leads to a 13% reduction (compared to that obtained with l_2_ = 150 mm) in the surface paint coat stripping efficiency. For the greatest spray length used of l_2_ = 400 mm, the reduction in treatment efficiency is just over 41%. The results from the use of a water-ice jet with pressure p_w_ = 35 MPa (Figure 13b) are similar to those obtained with lower pressure (p_w_ = 20 MPa).

The surface paint coat stripping efficiency as a function of the working jet length for PA2 aluminium alloy specimens using a water-ice jet created in a sprinkler head with length L_k_ = 200 mm is presented in Figure 14a,b. It was found that an increase in the working jet length from l_2_ = 150 mm to l_2_ = 200 mm is accompanied by an increase in the surface paint coat stripping efficiency caused by an increase in the impact area of the water-ice jet.

For a water-ice jet with pressure p_w_ = 20 MPa (Figure 14a), this increase is 9% on average. The maximum increase in treatment efficiency of less than 26% is obtained when the working jet length is changed from l_2_ = 150 mm to l_2_ = 250 mm. Further increase of the working length from l_2_ = 150 mm to l_2_ = 300 mm only results in a 9% increase in surface paint coat stripping efficiency. The use of l_2_ = 350 mm leads to a 15% reduction (compared to that obtained with l_2_ = 150 mm) in surface paint coat stripping efficiency. For the largest working jet length used l_2_ = 400 mm, the decrease in treatment efficiency is just over 41% (Figure 14a). The results from the use of a water-ice jet with pressure p_w_ = 35 MPa (Figure 14b) are similar to those obtained with lower pressure (p_w_ = 20 MPa).

In the paint stripping process, apart from dry ice output and the distance between the sprinkler head outlet, the spray angle is also of great importance (Figure 15a,b). This is connected both with the so-called jet reflection and the preservation of the mechanical properties of the coating during the treatment process. In natural conditions, the paint coating is characterised by elasticity, but during an interaction of a large number of CO_2_ dry ice particles with the temperature of 194.6 K, a local spot depression of its temperature occurs, which leads to the formation of thermal stress that causes microcracks, which initiate the erosion process [10]. The effect of the jet spray angle on the surface paint coat stripping efficiency from PA2 aluminium alloy when using a water-ice jet with pressure p_w_ = 20 MPa and p_w_ = 35 MPa is presented in Figure 15a,b.

With an increase of the jet spray angle κ = 30 ÷ 60°, there is an almost linear increase in surface paint coat stripping efficiency. Further increase of the spray angle up to κ = 90° causes an increase of the treatment efficiency yet that is not as sharp as observed with smaller spray angles. It can therefore be accepted that the best machining results are obtained with a spray angle of κ = 75 ÷ 90°.

The effect of water jet pressure on paint removal efficiency is shown in Figure 16. As the pressure increases, the processing efficiency increases. For example, increasing the pressure from p_w_ = 20 MPa to p_w_ = 25 MPa increased the treatment efficiency to ∆${\dot{Q}}_{F}$ = 0.014 m^2^·h^−1^ for the PA2 aluminum alloy. Further increase of the jet pressure to p_w_ = 30 MPa, with unchanged other processing parameters, resulted in paint removal efficiency of ${\dot{Q}}_{F}$ = 0.201 m^2^·h^−1^. This is an outgrowth of the treatment efficiency by ∆${\dot{Q}}_{F}$ = 0.012 m^2^·h^−1^ compared to that obtained for a jet with pressure p_w_ = 25 MPa.

## 4. Conclusions

The optical microscope by Kestler—Vision Engineering Dynascope Ltd. with the ND 1300 Quadra-Chek measuring system and the sensitivity of ±0.001 was used to measure the surface quality of the processed materials. Based on the research and analysis of the results, it was found that:The use of a high-pressure water-ice jet with maximum pressure p_w_ = 35 MPa does not cause any changes in the geometric structure of the surface of X5CrNi18-10 steel and PA2 aluminium alloy.In the case of brittle materials, such as polymethyl methacrylate PMMA, which form bases for lacquer coatings, it is necessary to use a water-ice jet with the maximum working pressure of p_w_ = 20 MPa and dry ice flow rate of ${\dot{m}}_{L}=156$ kg·h^−1^. The use of a water jet with pressure p_w_ = 35 MPa and dry ice flow rate ${\dot{m}}_{L}=208$ kg·h^−1^ causes traces of jet impact in the form of small chipping of the base material (PMMA polymethyl methacrylate).The highest surface treatment efficiency is obtained with a higher water jet pressure and an appropriately increased dry ice particle flow rate. For water jet pressure p_w_ = 20 MPa, the best results are obtained when dry ice flow rate is ${\dot{m}}_{L}=156÷182$ kg·h^−1^, while for water jet pressure p_w_ = 35 MPa, this output should not exceed ${\dot{m}}_{L}=234\;$ kg·h^−1^.In order to obtain the maximum paint coat stripping efficiency irrespective of the water jet pressure and the dry ice particle flow rate, the working length of the jet needs to be l_2_ = 250 mm and the spray angle needs to be κ = 75 ÷ 90°.Increasing the pressure of the water jet results in higher efficiency of paint removal. The increase of processing efficiency is proportional in the tested range of jet pressure p_w_ = 20 ÷ 35 MPa.The above research proves that one of the effective methods of removing paint coatings is the use of a high-pressure stream of water with ice. This method of processing allows for effective preparation of the substrate for the application of a renovation paint coating, while simultaneously maintaining appropriate processing parameters in that it does not change the geometric structure of the substrate.

## Figures and Tables

**Figure 1 materials-15-01168-f001:**
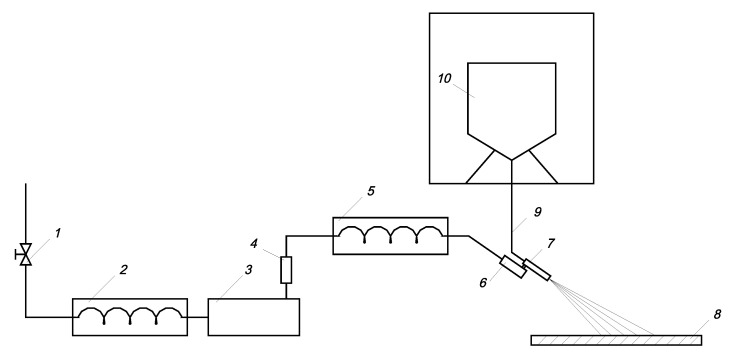
Scheme of the experimental system: 1—valve, 2—radiator, 3—water pump, 4—control system, 5—radiator, 6—high-pressure gun, 7—sprinkler, 8—work surface, 9—suction hose, 10—CO_2_ dry ice particle tank.

**Figure 2 materials-15-01168-f002:**
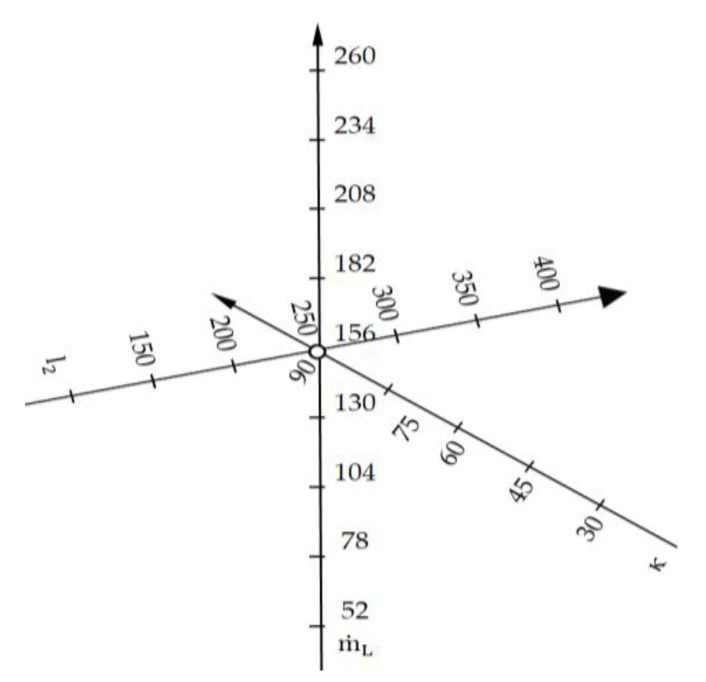
Graphical illustration of the test programme.

**Figure 3 materials-15-01168-f003:**
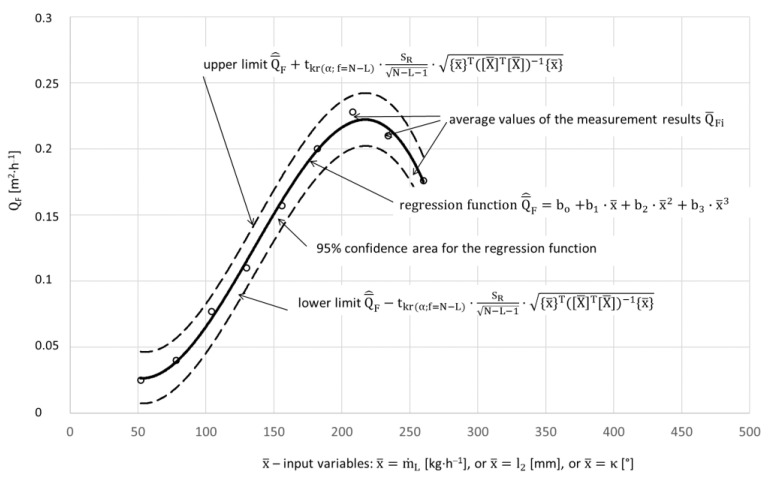
Illustration of the influence of factor x on the average paint removal efficiency, lower and upper limits, and the 95% confidence region for the regression function.

**Figure 4 materials-15-01168-f004:**
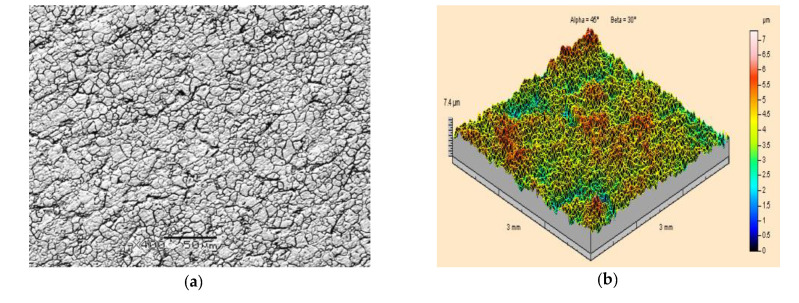
Surface topography of X5CrNi18-10 steel specimens prior to the application of paint coating and prior to ice-water jet treatment: (**a**) Grey scale map under scanning electron microscope (400×magnification); (**b**) Isometric image.

**Figure 5 materials-15-01168-f005:**
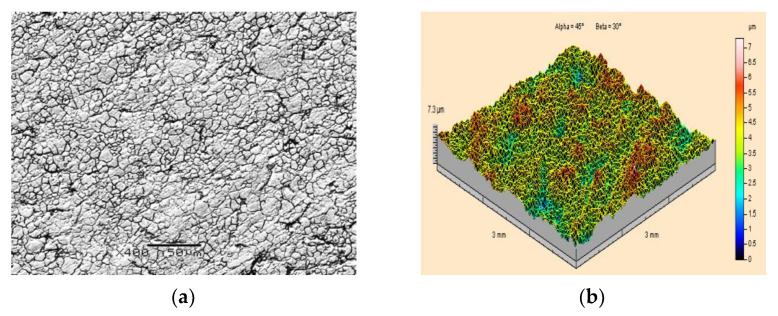
Surface topography of X5CrNi18-10 steel specimens after water-ice jet treatment (p_w_ = 35 MPa, ${\dot{m}}_{L}=208$ kg·h^−1^, l_2_ = 250, κ = 90°): (**a**) Grey scale map under a scanning electron microscope (400×magnification); (**b**) Isometric image.

**Figure 6 materials-15-01168-f006:**
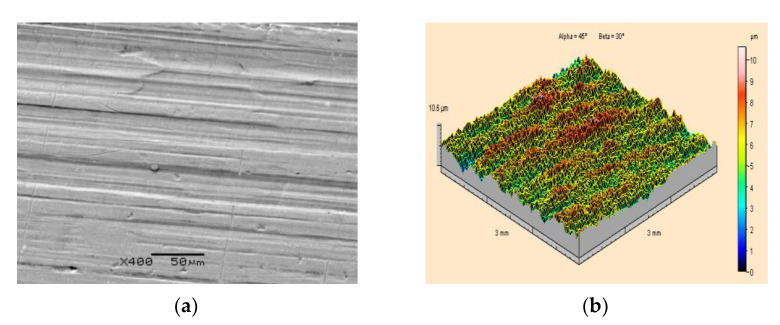
Surface topography of a PA2 aluminium alloy specimen prior to the application of paint coating and before high-pressure water-ice jet treatment: (**a**) Scanning electron microscope image (400×magnification); (**b**) Isometric image.

**Figure 7 materials-15-01168-f007:**
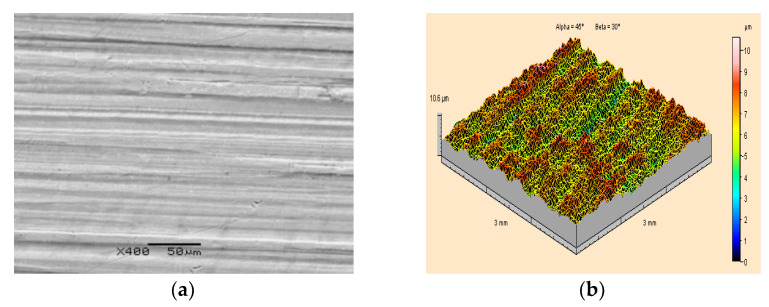
Surface topography of PA2 aluminium alloy specimens after ice-water jet treatment (p_w_ = 35 MPa, ${\dot{m}}_{L}=208$ kg·h^−1^, l_2_ = 250 mm, κ = 90°): (**a**) Scanning electron microscope image (400×magnification); (**b**) Isometric image.

**Figure 8 materials-15-01168-f008:**
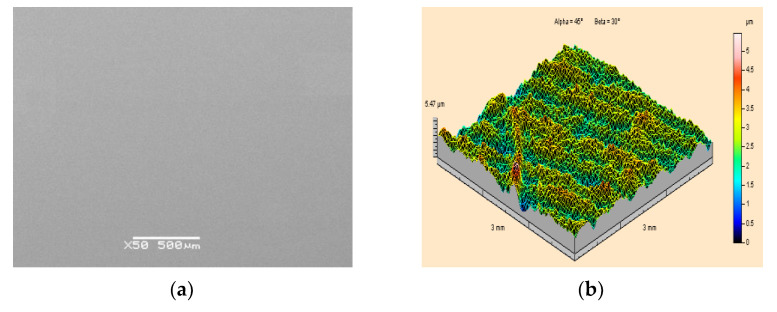
View of the surface of a polymethyl methacrylate PMMA specimen prior to the application of a paint coating and prior to high-pressure water-ice jet treatment: (**a**) Scanning electron microscope image (50×magnification); (**b**) Isometric image.

**Figure 9 materials-15-01168-f009:**
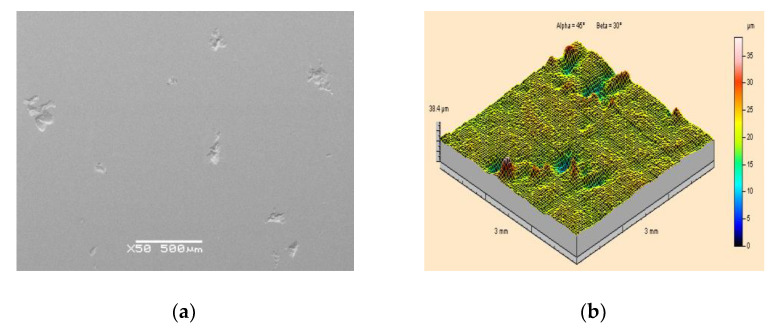
View of the surface of polymethyl methacrylate PMMA specimen after high-pressure water-ice jet treatment (p_w_ = 35 MPa, ${\dot{m}}_{L}=208$ kg·h^−1^, l_2_ = 250 mm, κ = 90°): (**a**) Scanning electron microscope image (50×magnification); (**b**) Isometric image.

**Figure 10 materials-15-01168-f010:**
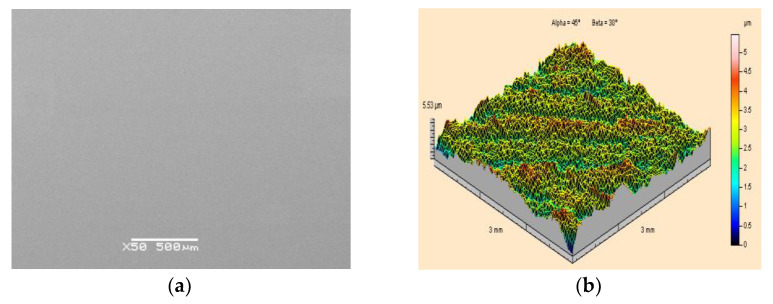
View of the surface of PMMA polymethyl methacrylate specimens after high-pressure water-ice jet treatment (p_w_ = 20 MPa,${\dot{m}}_{L}=156$ kg·h^−1^, l_2_ = 250 mm, κ = 90°): (**a**) Scanning electron microscope image (50×magnification); (**b**) Isometric image.

**Figure 11 materials-15-01168-f011:**
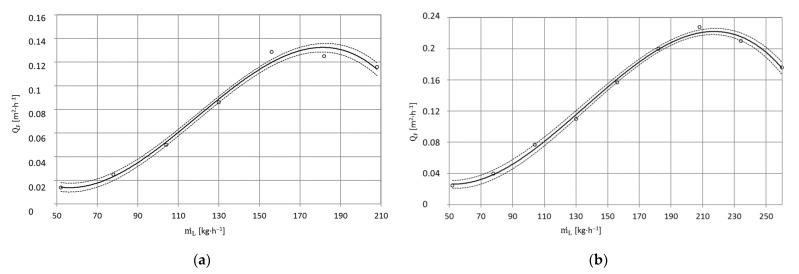
Influence of CO_2_ ${\dot{m}}_{L}$ [kg·h^−^^1^] dry ice output on the maximum surface efficiency $Q_{F}$ [m^2^·h^−^^1^] of paint stripping by high-pressure water-ice jet from X5CrNi18-10 steel surfaces (TS = 4 × 1.2 mm, L_k_ = 200 mm, l_2_ = 250 mm, κ = 90°): (**a**) jet pressure p_w_ = 20 MPa; (**b**) p_w_ = 35 MPa.

**Figure 12 materials-15-01168-f012:**
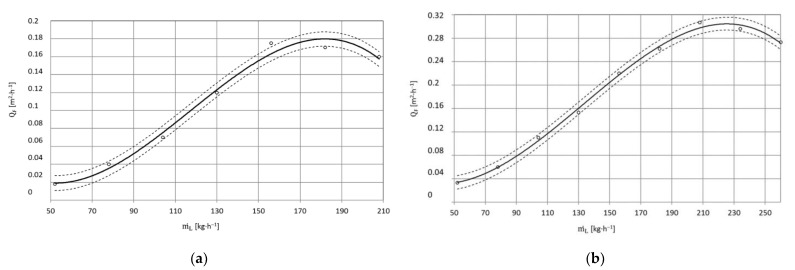
Influence of dry ice CO_2_ output ${\dot{m}}_{L}$ [kg·h^−1^] on maximum surface efficiency ${\bar{Q}}_{F}$ [m^2^·h^−^^1^] of paint coat stripping by high-pressure water-ice jet from PA2 aluminium alloy surface (TS = 4 × 1.2 mm, L_k_ = 200 mm, l_2_ = 250 mm, κ = 90°): (**a**) Jet pressure p_w_ = 20 MPa; (**b**) p_w_ = 35 MPa.

**Figure 13 materials-15-01168-f013:**
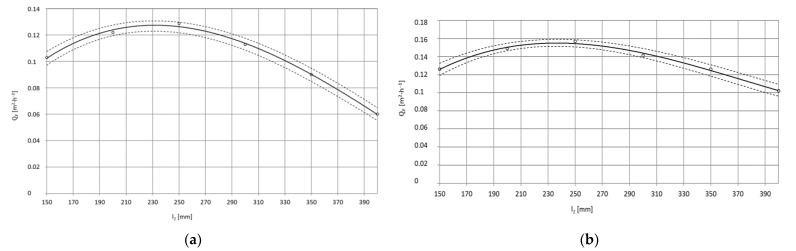
Influence of jet working length l_2_ [mm] on maximum surface efficiency ${\bar{Q}}_{F}$ [m^2^·h^−^^1^] of coat paint stripping with high-pressure water-ice jet from X5CrNi18-10 steel surface (TS = 4 × 1.2 mm, L_k_ = 200 mm, κ = 90°, ${\dot{m}}_{L}=156\;$ kg·h^−^^1^): (**a**) Jet pressure p_w_ = 20 MPa; (**b**) p_w_ = 35 MPa.

**Figure 14 materials-15-01168-f014:**
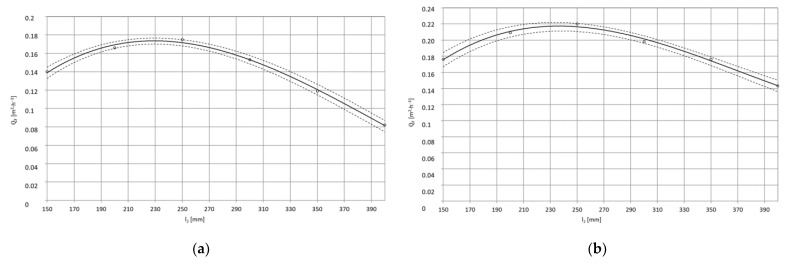
Influence of jet working length l_2_ [mm] on maximum surface efficiency $\;{\bar{Q}}_{F}\;$ [m^2^·h^−^^1^] of high-pressure water-ice jet paint coat stripping from PA2 aluminium alloy surface (TS = 4 × 1.2 mm, L_k_ = 200 mm, κ = 90°, ${\dot{m}}_{L}=156\;$ kg·h^−^^1^): (**a**) Jet pressure p_w_ = 20 MPa; (**b**) p_w_ = 35 MPa.

**Figure 15 materials-15-01168-f015:**
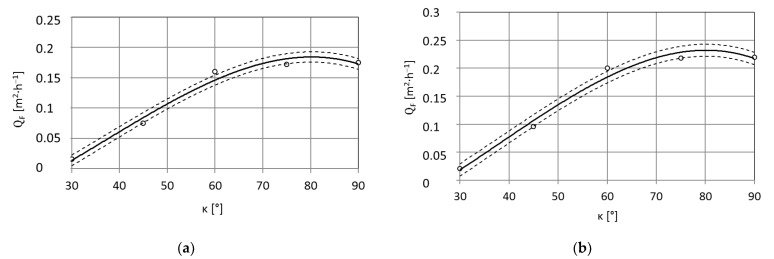
Impact of spray angle κ [°] on maximum surface efficiency ${\bar{Q}}_{F}\;$ [m^2^·h^−^^1^] of high-pressure water-ice jet paint coat stripping from PA2 aluminium alloy surface (TS = 4 × 1.2 mm, L_k_ = 200 mm, l_2_ = 250 mm, ${\dot{m}}_{L}=156\;$ kg·h^−^^1^): (**a**) Jet pressure p_w_ = 20 MPa; (**b**) p_w_ = 35 MPa.

**Figure 16 materials-15-01168-f016:**
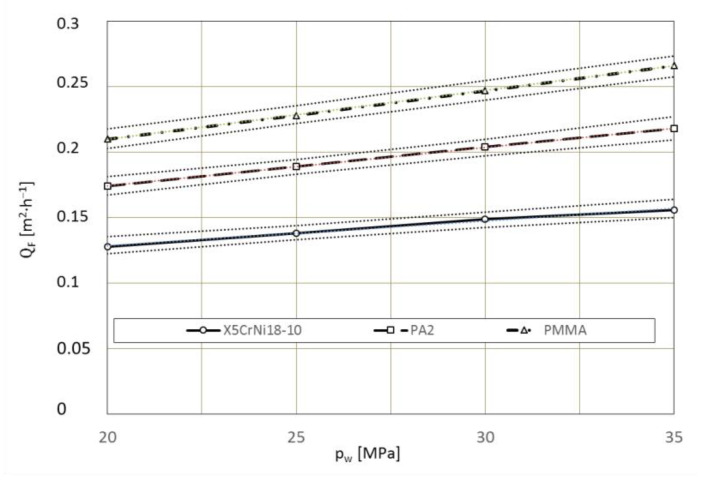
Influence of water-ice jet pressure p_w_ [MPa] on the maximum surface stripping efficiency $Q_{F}$ [m^2^·h^−^^1^] (TS = 4 × 1.2 mm, L_k_ = 200 mm, l_2_ = 250 mm, ${\dot{m}}_{L}=156\;$ kg·h^−^^1^, κ = 90°).

**Table 1 materials-15-01168-t001:** Mechanical and physical properties of materials used in the research.

Material Used in the Research	Material According to the Norm (EN)	Density ρ [kg × m^−ρ^]	Tensile Strength Rm [MPa]	Young’s Modulus E [GPa]	Yield Point Re [MPa]
X5CrNi18-10	1.4301	7900	540	200	230
PA2	AW-5251	2700	130	70	60
PMMA	-	1180	75	3	0.06

**Table 2 materials-15-01168-t002:** Sample test variants for p_w_ = 20 MPa, κ = 90°.

p_w_ [MPa]	20													
l_2_ [mm]	150							200						
${\dot{m}}_{L}$ [kg·h^−1^]	52	78	104	130	156	182	208	52	78	104	130	156	182	208
l_2_ [mm]	250							300						
${\dot{m}}_{L}$ [kg·h^−1^]	52	78	104	130	156	182	208	52	78	104	130	156	182	208
l_2_ [mm]	350							400						
${\dot{m}}_{L}$ [kg·h^−1^]	52	78	104	130	156	182	208	52	78	104	130	156	182	208

**Table 3 materials-15-01168-t003:** Sample test variants for p_w_ = 35 MPa, κ = 90°.

p_w_ [MPa]	35																	
l_2_ [mm]	150									200								
${\dot{m}}_{L}$ [kg·h^−1^]	52	78	104	130	156	182	208	234	260	52	78	104	130	156	182	208	234	260
l_2_ [mm]	250									300								
${\dot{m}}_{L}$ [kg·h^−1^]	52	78	104	130	156	182	208	234	260	52	78	104	130	156	182	208	234	260
l_2_ [mm]	350									400								
${\dot{m}}_{L}$ [kg·h^−1^]	52	78	104	130	156	182	208	234	260	52	78	104	130	156	182	208	234	260

**Table 4 materials-15-01168-t004:** Results of Sa and Sz measurements of X5CrNi18-10 steel samples before paint coating.

Sample No.	Sa [μm]	$\bar{z}$ [μm]	s(z) [μm]	R(z) [μm]	Sz [μm]	$\bar{z}$ [μm]	s(z) [μm]	R(z) [μm]
1	0.77	0.76	0.01	0.02	7.3	7.30	0.10	0.20
2	0.75				7.4			
3	0.76				7.2			
4	0.78	0.77	0.01	0.02	7.4	7.40	0.10	0.20
5	0.76				7.3			
6	0.78				7.5			
7	0.77	0.78	0.01	0.01	7.4	7.50	0.10	0.20
8	0.78				7.5			
9	0.78				7.6			
10	0.77	0.77	0.01	0.01	7.2	7.30	0.10	0.20
11	0.76				7.3			
12	0.77				7.4			
13	0.78	0.77	0.01	0.01	7.4	7.40	0.10	0.20
14	0.77				7.5			
15	0.77				7.3			
16	0.74	0.76	0.02	0.03	7.6	7.57	0.06	0.10
17	0.76				7.5			
18	0.77				7.6			
19	0.78	0.77	0.02	0.03	7.2	7.37	0.21	0.40
20	0.75				7.3			
21	0.77				7.6			
22	0.78	0.76	0.02	0.04	7.5	7.33	0.21	0.40
23	0.74				7.1			
24	0.76				7.4			
25	0.78	0.76	0.02	0.03	7.3	7.37	0.21	0.40
26	0.76				7.2			
27	0.75				7.6			
28	0.78	0.76	0.02	0.04	7.4	7.47	0.06	0.10
29	0.74				7.5			
30	0.76				7.5			
31	0.76	0.76	0.02	0.03	7.3	7.40	0.17	0.30
32	0.75				7.3			
33	0.78				7.6			
34	0.77	0.76	0.02	0.03	7.4	7.37	0.15	0.30
35	0.74				7.2			
36	0.77				7.5			
37	0.77	0.77	0.02	0.03	7.3	7.33	0.25	0.50
38	0.75				7.6			
39	0.78				7.1			
40	0.77	0.77	0.02	0.03	7.4	7.43	0.06	0.10
41	0.78				7.5			
42	0.75				7.4			
43	0.77	0.76	0.01	0.02	7.6	7.37	0.25	0.50
44	0.77				7.4			
45	0.75				7.1			
46	0.76	0.76	0.01	0.02	7.2	7.30	0.10	0.20
47	0.75				7.3			
48	0.77				7.4			

**Table 5 materials-15-01168-t005:** Results of Sa and Sz measurements of X5CrNi18-10 steel specimens after removal of paint coating (p_w_ = 20 MPa, l_2_ = 250 mm, κ = 90°).

Sample No.	$\dot{m_{L}}[\mathrm{kg}\cdot h{-}^{1}]$	Sa [μm]	$\bar{z}$ [μm]	s(z) [μm]	R(z) [μm]	Sz [μm]	$\bar{z}$ [μm]	s(z) [μm]	R(z) [μm]
1	52	0.77	0.76	0.01	0.02	7.4	7.37	0.06	0.10
2		0.76				7.4			
3		0.75				7.3			
4	78	0.77	0.77	0.01	0.01	7.3	7.37	0.21	0.40
5		0.77				7.2			
6		0.76				7.6			
7	104	0.77	0.77	0.02	0.03	7.3	7.50	0.17	0.30
8		0.79				7.6			
9		0.76				7.6			
10	130	0.77	0.76	0.02	0.03	7.3	7.33	0.15	0.30
11		0.74				7.2			
12		0.76				7.5			
13	156	0.77	0.77	0.01	0.01	7.4	7.43	0.06	0.10
14		0.77				7.4			
15		0.76				7.5			
16	182	0.75	0.76	0.02	0.03	7.5	7.50	0.10	0.20
17		0.75				7.4			
18		0.78				7.6			
19	208	0.75	0.75	0.01	0.01	7.3	7.40	0.10	0.20
20		0.75				7.4			
21		0.76				7.5			

**Table 6 materials-15-01168-t006:** Results of Sa and Sz measurements of X5CrNi18-10 steel specimens after removal of paint coating (p_w_ = 35 MPa, l_2_ = 250 mm, κ = 90°).

Sample No.	$\dot{m_{L}}[\mathrm{kg}\cdot h{-}^{1}]$	Sa [μm]	$\bar{z}$ [μm]	s(z) [μm]	R(z) [μm]	Sz [μm]	$\bar{z}$ [μm]	s(z) [μm]	R(z) [μm]
22	52	0.77	0.77	0.02	0.03	7.5	7.37	0.15	0.30
23		0.76				7.2			
24		0.79				7.4			
25	78	0.79	0.78	0.02	0.03	7.4	7.30	0.17	0.30
26		0.78				7.1			
27		0.76				7.4			
28	104	0.8	0.78	0.02	0.04	7.3	7.37	0.06	0.10
29		0.76				7.4			
30		0.78				7.4			
31	130	0.79	0.77	0.02	0.03	7.3	7.33	0.15	0.30
32		0.76				7.2			
33		0.77				7.5			
34	156	0.78	0.78	0.02	0.03	7.6	7.37	0.25	0.50
35		0.76				7.1			
36		0.79				7.4			
37	182	0.79	0.79	0.02	0.03	7.2	7.33	0.15	0.30
38		0.77				7.5			
39		0.8				7.3			
40	208	0.78	0.78	0.01	0.02	7.3	7.33	0.06	0.10
41		0.79				7.3			
42		0.77				7.4			
43	234	0.79	0.78	0.02	0.03	7.5	7.33	0.15	0.30
44		0.78				7.2			
45		0.76				7.3			
46	260	0.77	0.77	0.01	0.02	7.3	7.37	0.12	0.20
47		0.76				7.5			
48		0.78				7.3			

**Table 7 materials-15-01168-t007:** Results of Sa and Sz measurements of PA2 aluminum alloy samples before paint coating.

Sample No.	Sa [μm]	$\bar{z}$ [μm]	s(z) [μm]	R(z) [μm]	Sz [μm]	$\bar{z}$ [μm]	s(z) [μm]	R(z) [μm]
1	1.04	1.04	0.01	0.01	10.6	10.67	0.21	0.40
2	1.05				10.5			
3	1.04				10.9			
4	1.07	1.06	0.01	0.02	10.7	10.80	0.10	0.20
5	1.05				10.8			
6	1.06				10.9			
7	1.05	1.05	0.02	0.03	10.6	10.60	0.10	0.20
8	1.03				10.5			
9	1.06				10.7			
10	1.06	1.04	0.02	0.03	10.5	10.53	0.15	0.30
11	1.04				10.7			
12	1.03				10.4			
13	1.05	1.04	0.01	0.02	10.8	10.60	0.17	0.30
14	1.04				10.5			
15	1.03				10.5			
16	1.02	1.03	0.02	0.03	10.6	10.60	0.20	0.40
17	1.05				10.8			
18	1.03				10.4			
19	1.04	1.04	0.02	0.03	10.3	10.53	0.25	0.50
20	1.06				10.8			
21	1.03				10.5			
22	1.02	1.04	0.02	0.03	10.5	10.60	0.26	0.50
23	1.05				10.4			
24	1.04				10.9			
25	1.03	1.04	0.01	0.02	10.8	10.63	0.21	0.40
26	1.05				10.4			
27	1.05				10.7			
28	1.06	1.05	0.02	0.03	10.4	10.47	0.21	0.40
29	1.03				10.7			
30	1.05				10.3			
31	1.07	1.04	0.03	0.05	10.7	10.50	0.17	0.30
32	1.02				10.4			
33	1.04				10.4			
34	1.04	1.04	0.01	0.02	10.6	10.53	0.12	0.20
35	1.05				10.4			
36	1.03				10.6			
37	1.04	1.05	0.01	0.02	10.5	10.67	0.15	0.30
38	1.04				10.7			
39	1.06				10.8			
40	1.04	1.04	0.02	0.03	10.6	10.57	0.15	0.30
41	1.03				10.4			
42	1.06				10.7			
43	1.05	1.05	0.01	0.02	10.2	10.33	0.23	0.40
44	1.04				10.6			
45	1.06				10.2			
46	1.05	1.04	0.01	0.01	10.2	10.43	0.21	0.40
47	1.04				10.5			
48	1.04				10.6			

**Table 8 materials-15-01168-t008:** Results of Sa and Sz measurements of PA2 aluminum alloy samples before paint coating (p_w_ = 20 MPa, l_2_ = 250 mm, κ = 90°).

Sample No.	$\dot{m_{L}}[\mathrm{kg}\cdot h{-}^{1}]$	Sa [μm]	$\bar{z}$ [μm]	s(z) [μm]	R(z) [μm]	Sz [μm]	$\bar{z}$ [μm]	s(z) [μm]	R(z) [μm]
1	52	1.05	1.05	0.01	0.01	10.5	10.43	0.21	0.40
2		1.05				10.2			
3		1.06				10.6			
4	78	1.08	1.06	0.02	0.04	10.6	10.77	0.15	0.30
5		1.04				10.9			
6		1.06				10.8			
7	104	1.07	1.07	0.02	0.03	10.6	10.53	0.12	0.20
8		1.05				10.4			
9		1.08				10.6			
10	130	1.08	1.06	0.02	0.04	10.7	10.60	0.10	0.20
11		1.05				10.6			
12		1.04				10.5			
13	156	1.04	1.05	0.02	0.03	10.7	10.53	0.15	0.30
14		1.07				10.5			
15		1.05				10.4			
16	182	1.04	1.05	0.02	0.03	10.5	10.50	0.20	0.40
17		1.07				10.7			
18		1.04				10.3			
19	208	1.03	1.05	0.03	0.05	10.5	10.50	0.00	0.00
20		1.08				10.5			
21		1.04				10.5			

**Table 9 materials-15-01168-t009:** Results of Sa and Sz measurements of PA2 aluminum alloy specimens after removal of paint coating (p_w_ = 35 MPa, l_2_ = 250 mm, κ = 90°).

Sample No.	$\dot{m_{L}}[\mathrm{kg}\cdot h{-}^{1}]$	Sa [μm]	$\bar{z}$ [μm]	s(z) [μm]	R(z) [μm]	Sz [μm]	$\bar{z}$ [μm]	s(z) [μm]	R(z) [μm]
22	52	1.02	1.05	0.02	0.04	10.4	10.43	0.25	0.50
23		1.06				10.2			
24		1.06				10.7			
25	78	1.05	1.06	0.02	0.03	10.7	10.63	0.12	0.20
26		1.08				10.5			
27		1.06				10.7			
28	104	1.06	1.06	0.02	0.03	10.4	10.60	0.17	0.30
29		1.04				10.7			
30		1.07				10.7			
31	130	1.06	1.05	0.01	0.02	10.5	10.43	0.21	0.40
32		1.04				10.6			
33		1.05				10.2			
34	156	1.06	1.06	0.02	0.04	10.6	10.53	0.31	0.60
35		1.08				10.2			
36		1.04				10.8			
37	182	1.05	1.05	0.01	0.01	10.8	10.73	0.21	0.40
38		1.04				10.5			
39		1.05				10.9			
40	208	1.06	1.06	0.02	0.04	10.6	10.63	0.15	0.30
41		1.04				10.5			
42		1.08				10.8			
43	234	1.06	1.05	0.02	0.04	10.3	10.53	0.32	0.60
44		1.03				10.9			
45		1.07				10.4			
46	260	1.07	1.06	0.01	0.02	10.5	10.67	0.21	0.40
47		1.05				10.6			
48		1.06				10.9			

**Table 10 materials-15-01168-t010:** Sa and Sz measurement results of polymethyl methacrylate PMMA samples before paint coating.

Sample No.	Sa [μm]	$\bar{z}$ [μm]	s(z) [μm]	R(z) [μm]	Sz [μm]	$\bar{z}$ [μm]	s(z) [μm]	R(z) [μm]
1	0.48	0.48	0.01	0.02	5.48	5.48	0.07	0.13
2	0.47				5.42			
3	0.49				5.55			
4	0.5	0.48	0.02	0.03	5.49	5.49	0.08	0.15
5	0.48				5.42			
6	0.47				5.57			
7	0.48	0.49	0.01	0.02	5.43	5.49	0.07	0.13
8	0.5				5.56			
9	0.48				5.49			
10	0.5	0.49	0.02	0.03	5.48	5.47	0.06	0.12
11	0.47				5.41			
12	0.49				5.53			
13	0.49	0.50	0.01	0.02	5.47	5.48	0.02	0.04
14	0.5				5.5			
15	0.51				5.46			
16	0.49	0.49	0.01	0.02	5.49	5.46	0.03	0.06
17	0.5				5.43			
18	0.48				5.46			
19	0.49	0.48	0.01	0.02	5.47	5.47	0.04	0.07
20	0.47				5.43			
21	0.49				5.5			
22	0.51	0.50	0.02	0.03	5.49	5.46	0.03	0.06
23	0.5				5.47			
24	0.48				5.43			
25	0.49	0.48	0.01	0.02	5.46	5.46	0.03	0.05
26	0.47				5.44			
27	0.48				5.49			
28	0.5	0.49	0.02	0.03	5.51	5.48	0.03	0.05
29	0.49				5.48			
30	0.47				5.46			
31	0.49	0.49	0.01	0.02	5.57	5.49	0.07	0.14
32	0.48				5.43			
33	0.5				5.46			
34	0.49	0.49	0.02	0.03	5.5	5.47	0.03	0.06
35	0.5				5.47			
36	0.47				5.44			
37	0.47	0.49	0.02	0.03	5.47	5.48	0.03	0.05
38	0.5				5.46			
39	0.5				5.51			
40	0.49	0.49	0.01	0.01	5.47	5.47	0.04	0.07
41	0.5				5.5			
42	0.49				5.43			
43	0.49	0.49	0.02	0.04	5.44	5.47	0.04	0.07
44	0.47				5.51			
45	0.51				5.45			
46	0.49	0.50	0.01	0.02	5.41	5.43	0.02	0.04
47	0.51				5.43			
48	0.5				5.45			

**Table 11 materials-15-01168-t011:** Sa and Sz measurement results of polymethyl methacrylate PMMA samples after paint removal (p_w_ = 20 MPa, l_2_ = 250 mm, κ = 90°).

Sample No.	$\dot{m_{L}}[\mathrm{kg}\cdot h{-}^{1}]$	Sa [μm]	$\bar{z}$ [μm]	s(z) [μm]	R(z) [μm]	Sz [μm]	$\bar{z}$ [μm]	s(z) [μm]	R(z) [μm]
1	52	0.5	0.50	0.02	0.03	5.5	5.49	0.05	0.09
2		0.48				5.44			
3		0.51				5.53			
4	78	0.51	0.49	0.02	0.04	5.54	5.51	0.07	0.13
5		0.5				5.43			
6		0.47				5.56			
7	104	0.49	0.49	0.02	0.04	5.46	5.50	0.04	0.08
8		0.51				5.54			
9		0.47				5.49			
10	130	0.5	0.50	0.01	0.01	5.46	5.47	0.04	0.08
11		0.49				5.43			
12		0.5				5.51			
13	156	0.5	0.50	0.02	0.03	5.49	5.48	0.05	0.10
14		0.52				5.52			
15		0.49				5.42			
16	182	0.52	0.50	0.02	0.03	5.49	5.48	0.04	0.07
17		0.49				5.51			
18		0.49				5.44			
19	208	0.5	0.50	0.01	0.02	5.42	5.48	0.05	0.10
20		0.49				5.49			
21		0.51				5.52			

**Table 12 materials-15-01168-t012:** Sa and Sz measurement results of polymethyl methacrylate PMMA samples after paint removal (p_w_ = 35 MPa, l_2_ = 250 mm, κ = 90°).

Sample No.	$\dot{m_{L}}[\mathrm{kg}\cdot h{-}^{1}]$	Sa [μm]	$\bar{z}$ [μm]	s(z) [μm]	R(z) [μm]	Sz [μm]	$\bar{z}$ [μm]	s(z) [μm]	R(z) [μm]
22	52	1.45	1.44	0.02	0.04	38.7	38.40	0.26	0.50
23		1.42				38.2			
24		1.46				38.3			
25	78	1.48	1.45	0.04	0.07	38.3	38.43	0.32	0.60
26		1.47				38.8			
27		1.41				38.2			
28	104	1.5	1.47	0.04	0.07	38.8	38.43	0.32	0.60
29		1.48				38.2			
30		1.43				38.3			
31	130	1.51	1.47	0.05	0.09	38.6	38.37	0.21	0.40
32		1.42				38.3			
33		1.49				38.2			
34	156	1.49	1.49	0.02	0.03	38.1	38.40	0.26	0.50
35		1.5				38.6			
36		1.47				38.5			
37	182	1.49	1.50	0.01	0.02	38.6	38.50	0.26	0.50
38		1.51				38.7			
39		1.5				38.2			
40	208	1.48	1.48	0.04	0.07	38.4	38.40	0.20	0.40
41		1.44				38.6			
42		1.51				38.2			
43	234	1.5	1.48	0.03	0.06	38.2	38.43	0.32	0.60
44		1.44				38.8			
45		1.49				38.3			
46	260	1.52	1.50	0.02	0.03	38.3	38.40	0.26	0.50
47		1.5				38.7			
48		1.49				38.2			

**Table 13 materials-15-01168-t013:** Measuring results of average surface paint coat stripping efficiency ${\bar{Q}}_{F}$ [m^2^·h^−1^] from the surface of X5CrNi18-10 steel specimen as a function of the dry ice mass flow rate $\dot{m_{L}}$ [kg·h^−1^].

$\mathrm{Dry}\;\mathrm{ice}\;\mathrm{Mass}\;\mathrm{Flow}\;\mathrm{Rate}\;\dot{m_{L}}[\mathrm{kg}\cdot h{-}^{k}]$	Water Pressure p_w_ = 20 MPa p_w_ = 35 MPa Average Surface Efficiency ${\bar{Q}}_{F}\;[m^{2}\cdot h^{-h}]$	
52	0.014	0.025
78	0.025	0.040
104	0.050	0.077
130	0.086	0.110
156	0.129	0.157
182	0.125	0.200
208	0.116	0.228
234	-	0.210
260	-	0.176

**Table 14 materials-15-01168-t014:** Measuring results of average surface paint coat stripping efficiency ${\bar{Q}}_{F}$ [m^2^·h^−1^] from the surface of PA2.

$\mathrm{Dry}\;\mathrm{ice}\;\mathrm{Mass}\;\mathrm{Flow}\;\mathrm{Rate}\;\dot{m_{L}}[\mathrm{kg}\cdot h{-}^{k}]$	Water Pressure p_w_ = 20 MPa p_w_ = 35 MPa Average Surface Efficiency ${\bar{Q}}_{F}\;[m^{2}\cdot h^{-h}]$	
52	0.018	0.033
78	0.040	0.060
104	0.070	0.110
130	0.120	0.153
156	0.175	0.22
182	0.171	0.262
208	0.160	0.307
234	-	0.296
260	-	0.273

**Table 15 materials-15-01168-t015:** Measuring results of average surface paint coat stripping efficiency ${\bar{Q}}_{F}$ [m^2^·h^−1^] from the surface of specimen made of X5CrNi18-10 steel depending on jet working length $l_{2\;}$ [mm].

$\mathrm{Working}\;\mathrm{Jet}\;\mathrm{Length}\;l_{2}[\mathrm{mm}]$	Water Pressure p_w_ = 20 MPa p_w_ = 35 MPa Average Surface Efficiency ${\bar{Q}}_{F}\;[m^{2}\cdot h^{-h}]$	
150	0.103	0.126
200	0.122	0.149
250	0.129	0.157
300	0.113	0.141
350	0.090	0.126
400	0.060	0.102

**Table 16 materials-15-01168-t016:** Measuring results of average surface paint coat stripping efficiency ${\bar{Q}}_{F}$ [m^2^·h^−1^] of the PA2 aluminium alloy specimen depending on the working length of the spray $l_{2}$ [mm].

$\mathrm{Working}\;\mathrm{Jet}\;\mathrm{Length}l_{2}[\mathrm{mm}]$	Water Pressure p_w_ = 20 MPa p_w_ = 35 MPa Average Surface Efficiency ${\bar{Q}}_{F}\;[m^{2}\cdot h^{-h}]$	
150	0.140	0.176
200	0.166	0.209
250	0.175	0.220
300	0.153	0.198
350	0.119	0.176
400	0.082	0.143

**Table 17 materials-15-01168-t017:** Measuring results of average surface paint coat stripping efficiency ${\bar{Q}}_{F}$ [m^2^·h^−1^] of PA2 aluminium alloy specimen depending from spray angle κ [°].

Spray Angle κ [°]	Water Pressure p_w_ = 20 MPa p_w_ = 35 MPa Average Surface Efficiency ${\bar{Q}}_{F}\;[m^{2}\cdot h^{-h}]$	
30	0.016	0.021
45	0.075	0.096
60	0.160	0.200
75	0.172	0.218
90	0.175	0.220

**Table 18 materials-15-01168-t018:** Measuring results of average surface paint coat stripping efficiency ${\bar{Q}}_{F}$ [m^2^·h^−1^] for central values $\dot{m_{L}}$ = 156 [kg·h^-1^], $l_{2\;}$ = 250 [mm], κ = 90 [°].

Water Pressure p_w_ [MPa]	PMMA	PA2	X5CrNi18-10
	Average Surface Efficiency ${\bar{Q}}_{F}[m^{2}\cdot h{-}^{1}]$		
20	0.210	0.175	0.129
25	0.228	0.189	0.138
30	0.250	0.201	0.150
35	0.264	0.220	0.157

**Table 19 materials-15-01168-t019:** Matrix forms $\left[\bar{X}\right]$ for N = 5 and L = 4; N = 6 and L = 4; N = 7 and L = 4 as well as N = 9 and L = 4.

	Matrix Forms $\left[\bar{X}\right]$			
N = 5 and L = 4	1	30	900	27,000
	1	45	2025	91,125
	1	60	3600	216,000
	1	75	5625	421,875
	1	90	8100	729,000
N = 6 and L = 4	1	150	22,500	3,375,000
	1	200	40,000	8,000,000
	1	250	62,500	1,562,500
	1	300	90,000	27,000,000
	1	350	122,500	4,2875,000
	1	400	160,000	64,000,000
N = 7 and L = 4	1	52	2704	140,608
	1	78	6084	474,552
	1	104	10,816	1,124,864
	1	130	16,900	2,197,000
	1	156	24,336	3,796,416
	1	182	33,124	602,8568
	1	208	43,264	8,998,912
N = 9 and L = 4	1	52	2704	140,608
	1	78	6084	47,4552
	1	104	10,816	1,124,864
	1	130	16,900	2,197,000
	1	156	24,336	3,796,416
	1	182	33,124	6,028,568
	1	208	43,264	8,998,912
	1	234	54,756	12,812,904
	1	260	67,600	17,576,000

## Data Availability

Data sharing is not applicable to this article.

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
