# Peer review of "The Use of a High-Pressure Water-Ice Jet for Removing Worn Paint Coating in Renovation Process"

_materials, 2022, doi:10.3390/ma15031168_

Round 1

Reviewer 1 Report

The authors presents the results of investigations into the possibility of using a high-pressure water-ice jet as a new method for removing a worn-out paint coating from the surface of metal parts (including those found in the means of transport) and for preparing the base surface for the application of the renovation paint coating. Experimental investigations were carried out in four stages, on flat specimens sized: S×H=75×115 mm cut from a sheet metal made of various materials: steel X5CrNi18-10, PA2 aluminium alloy and PMMA polymethyl methacrylate (plastic). In the first stage, the surfaces of the specimens were subjected to observation by means of 3D surface topography under a scanning microscope; surface geometric structure (SGS) parameters were measured using a profilometer. The paper should be carefully revised. It may be accepted after major revision.

The mechanical and physical properties of the used materials should be listed in a table.

The novelty of the paper should be discussed in the introduction.

The difference between your paper and other published papers should be highlighted. Please check “optimization of abrasive water jet machining of sic reinforced aluminum alloy based metal matrix composites using taguchi–dear technique” which is published in the same journal, I e, MATERIALS.

The conclusion section is very weak, it should be rewritten.

A schematic diagram of the experimental set up should be included.

The quality of grey scale figures should be enhanced.

Why does the small length of a working jet result in high erosivity?

How did the authors select the cutting variables?

The design of experiments methodology should be described.

Reviewer 2 Report

Specific comments regarding the manuscript are given in form of comments (notes) in the pdf file included in this review. The general comments are given in the following:

The research conducted in this work is interesting for the field and could be useful, but the paper should organized much better so the reader should easily follow it. For these kinds of works I propose to use a clasical structer of the paper, Introduction, Experiments and methods, Results and Discussion, Conclusions.

Literature review is not sufficient, the state of the art technologies in this filed should be addressed in more detail in the introduction part. The comparison of high-pressure water ice jet with other technologies should be much more exhaustive. The advantages and potential in use of high-pressure  water ice are not clear nor the need for further research in this field. A lot of statements in the introduction are not supported by appropriate literature citing. At the end of the Introduction section a motivation for conducting a research and short explanation about what has been done in the work is required.

In this manuscript it is difficult to get the whole picture about the experimental work undertaken. The presentation of the conducted experimental work should be concentrated in one section with all its details. Use of tables for the samples designations, testing variants and statistical analysis greatly improve the presentation quality. Given that the statistical tool employed in this study is not appropriately set the design of experiment and statistical analysis employed should be reviewed.

It is not clear in the paper how many repetitions did you make in topography measurements. It is also not clear how did you prove that there is no remnant of paint on the evaluated surfaces.

For the journals of this rank, too much theoretical basics of used methods are not needed. The theory behind the surface roughness measurements and statistics are well known in material science.

The interpretations of the results include a lot of speculations not based on the results, nor on the literature, these should be avoided as much as possible. It cannot be said much more than results speak to themself.

The quality of the results discussion is mostly on a low level, it can be said that it almost does not exist. The English language should be on higher level. A lot of technical and other phrases used in this paper is very strange and difficult to perceive in a proper way.

Reviewer 3 Report

  1. Pg3 : Check the hardness value: 'The hardness of dry ice particles is estimated at 2÷3'
  2. Pg3 A JOEL JSM-5500LV scanning microscope… ‘ correct: scanning electron microscope (SEM)
  3. Adding Scale bar and correct the name of the instrument

Fig. 3a  add scale bar; correction: scanning electron microscope (not scanning microscope) or SEM

Fig. 3b 3nm scale bar looks not right

Fig. 4a Scale bar; correction: scanning electron microscope (not scanning microscope) or SEM

Fig. 4b 3nm scale bar looks not right

Fig 5 & Fig 6: correction as above Fig 3&4

4.All SEM figures (Fig 3-Fig6) should have a high magnification (e.g. x400) and low magnification (e.g x50)

  1. More than one area of roughness measurement should be carried out for statistically reliable results

6.Pg 18

Correction: ÷ in ‘With an increase of the jet spray angle =30÷60

Correction: ÷ in ‘spray angle of =75÷90.’

  1. More references are needed especially in Section 4

,

Reviewer 4 Report

Article “The use of a high-pressure water-ice jet for removing worn paint coating in renovation process”

The article deals with a topic that is constantly topical. The article is in-depth and covers the issues in sufficient detail. The authors have dealt with the subject in a way that shows they are well versed in the subject. In my view, the article is ready to be published as such.  

Round 2

Reviewer 1 Report

The manuscript is ready for publication.

Author Response

Please find attached the current version of the article, taking into account the suggestions of all reviewers

Reviewer 2 Report

Authors of the paper replayed only to my general comments, and they avoided answering the specific comments that I gave in the enclosed PDF file (maybe they did not receive the PDF). I spent a lot of time to give those specific comments (about 60) in PDF and raised a lot of specific questions about this work. Given that the paper has numerous flaws only answering to these specific comments could lead to the improvement of the whole manuscript. Therefore, I don`t see the improvement of this manuscript at all.

The answers to my general comments just partly improved the manuscript, these were not skillfully implemented and therefore these failed to answer to all problems in manuscript.

I have to say that it is really inconvenient to refuse to make changes that reviewer asked to make, instead authors argue or just disregarded to make changes. Which these authors do. Their rebuttal is based on self opinions and on not verifiable facts from the literature. I think that such communication is not appropriate for scientific paper review process.

Authors refuse to understand that detailed explanation of theory behind standard surface roughness parameters and standard statistical analysis does not have be included in such manuscript.

Statistical analysis (regression analysis) is not set appropriately set, it do not include all relevant parameters and therefore these results does not have sufficient scientific value.

I stick to my previous opinion regarding this paper, it has serious flaws and should not be published in present form in journal of such rank that Journal Materials belongs to.
